# The Biological Role of Apurinic/Apyrimidinic Endonuclease1/Redox Factor-1 as a Therapeutic Target for Vascular Inflammation and as a Serologic Biomarker

**DOI:** 10.3390/biomedicines8030057

**Published:** 2020-03-10

**Authors:** Yu Ran Lee, Hee Kyoung Joo, Byeong Hwa Jeon

**Affiliations:** 1Research Institute for Medical Sciences, College of Medicine, Chungnam National University, 266 Munhwa-ro, Jung-gu, Daejeon 35015, Korea; lyr0913@cnu.ac.kr (Y.R.L.); hkjoo79@cnu.ac.kr (H.K.J.); 2Department of Physiology, College of Medicine, Chungnam National University, 266 Munhwa-ro, Jung-gu, Daejeon 35015, Korea

**Keywords:** endothelial dysfunction, vascular inflammation, APE1/Ref-1, cardiovascular diseases, subcellular localization, serological biomarkers

## Abstract

Endothelial dysfunction promotes vascular inflammation by inducing the production of reactive oxygen species and adhesion molecules. Vascular inflammation plays a key role in the pathogenesis of vascular diseases and atherosclerotic disorders. However, whether there is an endogenous system that can participate in circulating immune surveillance or managing a balance in homeostasis is unclear. Apurinic/apyrimidinic endonuclease 1/redox factor-1 (henceforth referred to as APE1/Ref-1) is a multifunctional protein that can be secreted from cells. It functions as an apurinic/apyrimidinic endonuclease in the DNA base repair pathway and modulates redox status and several types of transcriptional factors, in addition to its anti-inflammatory activity. Recently, it was reported that the secretion of APE1/Ref-1 into the extracellular medium of cultured cells or its presence in the plasma can act as a serological biomarker for certain disorders. In this review, we summarize the possible biological functions of APE1/Ref-1 according to its subcellular localization or its extracellular secretions, as therapeutic targets for vascular inflammation and as a serologic biomarker.

## 1. Endothelial Dysfunction and Vascular Inflammation

Endothelial cell activation or dysfunction is defined by the endothelial expression of cell-surface adhesion molecules. The expression of adhesion molecules and the subsequent monocyte adhesion are considered as early events in the development of atherosclerosis [1]. Vascular inflammation plays a key role in the pathogenesis of vascular diseases and atherosclerotic disorders [2]. The inflammatory reaction is a series of complex interactions between inflammatory cells or stimuli and defense cells, such as macrophages and endothelial cells [3]. This interactive reaction triggers an inflammatory response in vascular cells by the activating of increased proinflammatory mediators and/or molecules, and cytokines [4].

This type of interactive reaction helps to eliminate the initial cause of injury, clear out inflammatory foci or cells, and helps the host cells to survive. The adhesion of leukocytes to the vascular endothelium is a hallmark of the inflammatory process [5]. Several types of antiadhesion therapeutic molecules are being developed for inflammatory diseases [6]. Adhesion molecules such as intercellular adhesion molecule 1 (ICAM-1), vascular cell adhesion molecule 1 (VCAM-1), and platelet endothelial cell adhesion molecule, are involved in the recruitment of monocytes/macrophages to the inflamed sites in the vascular tissue [7]. The expression of cell adhesion molecules, such as VCAM-1, represents one of earliest pathological changes in vascular inflammation diseases such as atherosclerosis [2]. Atherosclerosis is a chronic inflammatory disease of the vascular tissue that is largely driven by an innate immune response from the macrophages [8]. Atherosclerosis is characterized by lipid accumulation and inflammatory infiltration of the arterial walls [9]. The accumulation of a lipid plaque and lipid-forming macrophage foam cells in the intima of the inflamed artery has been recognized as a hallmark of atherosclerosis [10]. Macrophages actively contribute in vascular inflammation by secreting proinflammatory cytokines, such as tumor necrosis factor (TNF)-alpha [11]. There is increasing evidence that TNF-blocking agents including TNF receptor blockade have successfully been used to treat systemic inflammatory disorders, such as rheumatoid arthritis [12]. A recent interesting study evaluated the inhibition of inflammatory cytokines for treating atherothrombosis [13], suggesting that cytokine inhibition can help resolve inflammation and maintain homeostasis, and is thus is crucial for atheroprotection. Because cholesterol is a key component of arterial plaques, a detailed understanding of the cholesterol transport system can lead to approaches that help to lower the risk of atherosclerosis. Intracellular cholesterol can be exported through cholesterol transporters. Macrophage cholesterol efflux depends on the ATP-binding cassette transporters ABCA1 or ABCG1 [14]. The combined efficiency of ABCA1 and ABCG1 promotes foam cell accumulation by inhibiting macrophage cholesterol efflux and accelerates atherosclerosis in mice [15,16] suggesting a target for atherosclerotic cardiovascular diseases. A new target molecule capable of efficiently monitoring vascular inflammation, extracellularly secreted as needed to act as a biomarker, and able to control vascular inflammation including sepsis or cytokine storms, is required. Here, we introduce APE1/Ref-1 as a potential new target capable of meeting these demands.

## 2. APE1/Ref-1 Protein Has Several Cellular Functions

Is there an endogenous system that can participate in circulating immune surveillance or managing the balance in homeostasis? The molecule that can act in circulatory surveillance is a functional protein, which can recognize the DNA damage, and is sensitive to their redox status and their existence in the biological fluids. To date, the cellular localization of APE1/Ref-1 exhibits three types—nuclear, cytoplasmic/mitochondrial, and secretory. Under basal conditions, APE1/Ref-1 is localized in the nucleus, and its localization is dynamically regulated, resulting in its cytoplasmic/mitochondrial translocation or extracellular secretion [17]. Overexpression of APE1/Ref-1 is inhibited by TNF-α-induced endothelial cell activation in cultured endothelial cells [18]. In contrast, heterozygous APE1/Ref-1 (+/−) mice showed endothelial dysfunction and hypertension [19], suggesting an important role for APE1/Ref-1 in endothelial functions. Conventional knockout of APE1/Ref-1 causes early embryonic lethality on embryonic day 5 to E9 [20,21]. Therefore, it is difficult to evaluate the biological function or phenotype changes in homozygous APE1/Ref-1-knockout mice. A recent study showed that secretory APE1/Ref-1 inhibited proinflammatory cytokines and inflammation in lipopolysaccharide-treated mice [22]. For approximately 20 decades, extranuclear functions in systemic inflammation and endothelial activation as well as basic nuclear functions in DNA basic repair and genomic stability have been revealed (Figure 1).

### 2.1. Nuclear Function of APE1/Ref-1

The primary subcellular localization of APE1/Ref-1 is in the nucleus in most cells or tissues [24]. This appears to be because of its fundamental activity in the base excision repair pathway of DNA lesions. APE1/Ref-1 hydrolyzes the DNA adjacent to the 5′-end of an apurinic/apyrimidinic site to produce a nick with a 3′-hydroxyl group and a 5′-deoxyribose phosphate group like a skilled nucleic acid surgeon [25]. The APE1/Ref-1-deficient cells show hypersensitivity to DNA damaging agents [26,27]. APE1/Ref-1 also regulates the redox activity of several transcription factors such as activator protein-1 (AP-1) and nuclear factor kappa B (NF-κB) [17]. The formation of disulfide bonds in APE1/Ref-1 is important in redox activity with cysteine residues C65 and C93 playing key roles in the thiol-mediated redox reactions [28,29]. The calcification of vascular smooth muscle cells is strongly correlated with intracellular ROS production and apoptosis [30]. Recently, Lee et al. showed that the redox function of APE1/Ref-1 prevents inorganic phosphate-induced calcification of vascular smooth muscle cells by inhibiting oxidative stress and osteoblastic differentiation [31]. As the overexpression of APE1/Ref-1 inhibits endothelial apoptosis, the redox-sensitive APE1/Ref-1 plays a critical role in endothelial cell survival in response to inflammatory cytokines including tumor necrosis factor-alpha [32].

### 2.2. Cytoplasmic Function of APE1/Ref-1

APE1/Ref-1 has also been detected in other areas in addition to the nucleus; cytoplasmic and mitochondrial APE1/Ref-1 have also been reported [33,34]. Cytoplasmic overexpression of APE1/Ref-1 is attenuated by the upregulation of high-mobility group box1 (HMGB-1)-mediated ROS generation, cytokine secretion, and cyclooxygenase-2 expression in macrophage cells [35]. S-Nitrosoglutathion (GSNO), a nitric oxide donor, induces the nuclear export of APE1/Ref-1 in a chromosome-region maintenance-1 (exportin-1)-independent manner [36]. This nuclear-cytoplasmic translocation of APE1/Ref-1 is dependent on the nitrosation at the target sites Cys93 and Cys310 in APE1/Ref-1. The N-terminal of 20 amino acids of APE1/Ref-1 includes the nuclear localization signal, as the cytoplasmic proportion of APE1/Ref-1 increased with the deletion of the N-terminal of 20–35 amino acids [19,29,36]. APE1/Ref-1 also contains a potential nuclear export sequences (NES) at amino acids 64-80. A deletion mutant of APE1/Ref-1 (60–80) showed a slight interference with cell viability, suggesting the important role of the cytoplasmic localization of APE1/Ref-1 in cell viability [36]. Additionally, the cytoplasmic expression of APE1/Ref-1 has antioxidant and anti-inflammatory functions in astrocyte or endothelial cells [29,37]. Hypoxia resulted in a significant decrease in APE1/Ref-1 expression in human umbilical vein endothelial cells [38]. A novel extranuclear function of APE1/Ref-1 in endothelial oxidative stress and apoptosis is that it protects against hypoxia-reoxygenation-induced apoptosis by modulating cytoplasmic rac-1-regulated ROS generation [39]. Recently, Hao et al. reported that APE1/Ref-1 overexpression inhibited hypoxia-reoxygenation, which induced an increase in ROS and NADPH oxidase expression and inhibited the mitochondrial dysfunction in H9c2 cardiomyocytes [38].

Endothelial mitochondria are a critical target of oxidative stress and DNA damage, and thus play a crucial role in the signaling during cellular responses [40]. Phorbol 12-myristate 13-acetate (PMA), an activator of protein kinase C, induces ROS generation and increases mitochondrial translocation of APE1/Ref-1 [41]. Moreover, the overexpression of APE1/Ref-1 suppresses PMA-induced mitochondrial dysfunction. In contrast, the gene silencing of APE1/Ref-1 increases the sensitivity of mitochondrial dysfunction, suggesting that the mitochondrial APE1/Ref-1 contributes to the protective role of protein kinase C-induced mitochondrial dysfunction in endothelial cells [41]. Mitochondrial APE1/Ref-1 is also involved in repairing mitochondrial DNA lesions caused by oxidative and alkylating agents [42]. APE1/Ref-1 interacts with the mitochondrial import and assembly protein Mia40, which is responsible for APE1/Ref-1 trafficking into the mitochondria [42]. A recent study using haploinsufficient APE1/Ref-1 mice revealed slower repair kinetics of azoxymethane-induced mitochondrial DNA damage, suggesting that APE1/Ref-1 is important for preventing changes in mitochondrial DNA integrity during azoxymethane-induced colorectal cancer [43].

### 2.3. Extracellular Function of APE1/Ref-1

Mammalian cells may secrete several types of cellular proteins. In 2013, the secretion of APE1/Ref-1 into the cultured medium in response to hyperacetylation [44] and the presence of plasma APE1/Ref-1 in lipopolysaccharide-induced endotoxemic mice were first reported [45]. Thus, secreted APE1/Ref-1 protein likely has a distinct function. It is thought that the fundamental function of an intracellular protein is performed even when the protein is secreted from the cells. The cysteine residues of APE1/Ref-1 have a reducing activity for the redox regulation of target proteins [46]. Nath et al. reported that the extracellular APE1/Ref-1 induces the production and secretion of the proinflammatory cytokine IL-6 and extracellular APE1/Ref-1 treatment activates the transcriptional factor NF-κB [47]. In contrast, the anti-inflammatory activities of secreted APE1/Ref-1 have been reported, which is thought to be exerted by the reducing activity of APE1/Ref-1 via thiol exchanges in the extracellular domain of cytokine receptors [48]. Recently, Joo et al. demonstrated the in vivo activity of extracellularly secreted APE1/Ref-1, which exerts inhibitory effects on lipopolysaccharide (LPS)-induced inflammation and has a potential for treating LPS-induced endotoxemia or systemic inflammation such as cytokine storms [22]. Under endotoxemic conditions, multiple organ failure is caused by uncontrolled inflammatory responses such as cytokine storms or cytokine overproduction [49]. Interestingly, the secreted APE1/Ref-1 inhibited the LPS-induced proinflammatory mediators such as TNF-α, IL-1β, and IL-6, and chemotactic cytokines such as monocyte chemoattractant protein-1 (MCP-1), suggesting that the secretory APE1/Ref-1 inhibits LPS-induced cytokine production [22]. Reports of the extracellular secretions of APE1/Ref-1 have shown consistent results but have not agreed on the extracellular functions. Taken together, the anti-inflammatory effects of secretory APE1/Ref-1 in vivo as well as the therapeutic potential of recombinant APE1/Ref-1 protein in endotoxemic or inflammatory conditions have been suggested (Figure 2). The diverse biological functions of APE1/Ref-1 according to its subcellular localization are summarized in Table 1.

## 3. Mechanism of APE1/Ref-1 Secretion

There are two possible mechanisms for the extracellular secretion of the APE1/Ref-1 protein—active secretion and passive release. APE1/Ref-1 is actively secreted by inflammatory cells such as macrophages or monocytes and endothelial cells in response to hyperacetylation signals [44]. However, different exogenous stimuli such as trichostatin A, LPS, testosterone, and coxsackievirus B3 can induce the secretion of APE1/Ref-1 [45,55,56,57]. Intracellular hyperacetylation conditions may be important intracellular signals for the secretion of APE1/Ref-1 in normal or tumor cells [48,54,58].

Until now, this active secretion of APE1/Ref-1 has been known to be initiated by transporter and vesicle formation; it is mediated by a nonclassical transport pathway (Figure 3). As evidence of this, brefeldin A, an inhibitor of the endoplasmic reticulum-to-Golgi classical transport pathway, did not affect APE1/Ref-1 secretion [57]. Active secretion of APE1/Ref-1 is not be involved in the classical endoplasmic reticulum-to-Golgi complex secretory pathway because of the absence of a leader peptide sequence. Trichostatin A-mediated acetylation was shown to cause post-translational modification of APE1/Ref-1 (including Lys 6 and Lys 7 of APE1/Ref-1) [59]. This acetylation reduces the net charge and increases the hydrophobicity of APE1/Ref-1, leading to cytoplasmic localization and secretion. Additionally, trichostatin A did not induce the secretion of lysine-mutated APE1/Ref-1 (K6R/K7R) [44]. Pharmacological inhibition by probenecid and glyburide on acetylation-induced APE1/Ref-1 secretion suggested the possible involvement of ABC transporters [57]. In a human monocyte cell line, APE1/Ref-1 was secreted from the monocytes upon inflammatory challenges via extracellular vesicle-mediated secretion pathways [47]. There is an interesting report describing vesicle formation in the release of APE1/Ref-1 in breast tumor cell lines. Hyperacetylated MDA-MD-231 cells, which were stimulated with aspirin, released vesicles containing APE1/Ref-1 according to analysis using gold particle-labelled APE1/Ref-1 [54]. Further research is required to determine the molecular mechanism of APE1/Ref-1 secretion and if this mechanism is dependent on the cell type or endogenous stimuli. Extracellular APE1/Ref-1 may be passively released following endogenous cell damage or from necrotic cells. In necrotic or apoptotic cells, APE1/Ref-1 may be released into the cultured medium from the cytoplasm or nucleus, like HMGB-1 [60]. Therefore, the secreted APE1/Ref-1 in the extracellular milieu may be considered as a cell death marker and/or a serologic biomarker of certain disorders.

## 4. Extracellular APE1/Ref-1 as a Serological Biomarker

Since the concept of APE1/Ref-1 secretion was established in 2013, several studies have demonstrated the usefulness of APE1/Ref-1 as a serological biomarker for cardiovascular disorders and tumors (Table 2). Park et al. first reported APE1/Ref-1 in the plasma of endotoxemic rats as a 37 kDa immunoreactive band, suggesting that plasma APE1/Ref-1 is a useful biomarker for endotoxemia [45]. Jin et al. found that serum APE1/Ref-1 levels were elevated in the patients with coronary artery disease and were higher in myocadiac infarction than in angina in a study of clinical biomarkers [61]. Myocarditis is an inflammatory disease of the myocardium that causes cardiogenic shock, heart failure, and sudden death [62]. Myocarditis can only be diagnosed by endomyocardial biopsy [63]. Jin et al. reported that serum APE1/Ref-1 was elevated in experimental murine myocarditis; compared to N-terminal pro-B-type natriuretic peptide and troponin I, serum APE1/Ref-1 was more closely related to myocardial inflammation, reflecting the severity of myocardial injury in viral myocarditis without endomyocardial biopsy [55].

Vascular inflammation in the tumor microenvironment is associated with tumor angiogenesis or tumor metastasis [64]. In cancer research, the changes in the intracellular localization of APE1/Ref-1 in tissues have gained attention, as they are related to the prognosis of certain tumors. Overexpression of APE1/Ref-1 that is observed in tumor cells is associated with drug resistance of anticancer drugs and poor survival [65]. Moreover, gene silencing or the inhibition of redox activity of APE1/Ref-1 results in reduced drug resistance to anticancer drugs [66]. Therefore, APE1/Ref-1 is a target protein for tumor treatment. Recently, the usefulness of APE1/Ref-1 as a biomarker in various cancers has been demonstrated. Plasma or urine APE1/Ref-1 levels are significantly increased in patients with bladder cancer; area under the curve analysis revealed the diagnostic values of APE1/Ref-1 with high specificity and sensitivity [67,68]. There is increasing evidence for the role of serum APE1/Ref-1 as a new diagnostic biomarker for hepatocellular carcinoma [69], renal cell carcinoma and hepatobiliary carcinoma [70], cholangiocarcinoma [71], non-small cell lung cancer [72], and gastric cancer [73] as shown in Table 2.

## 5. Conclusions

In conclusion, APE1/Ref-1 has several cellular functions with an important role in DNA repair and redox regulation. In addition to the intracellular function of APE1/Ref-1, its extracellular function should be evaluated to develop therapeutic strategies. Recombinant APE1/Ref-1 protein including modifications may be used for circulating homeostatic surveillance, alone or in combination with treatment regimens, against endothelial dysfunction, inflammatory disorders or sepsis. Proteomic analysis of post-translational modifications of the APE1/Ref-1 protein in biological samples would improve the understanding of the diversity of APE1/Ref-1 function. Clinical studies of APE1/Ref-1 analysis as a biomarker in human samples will help in the diagnosis and follow-up of cardiovascular disorders, including coronary artery disease.

## Figures and Tables

**Figure 1 biomedicines-08-00057-f001:**
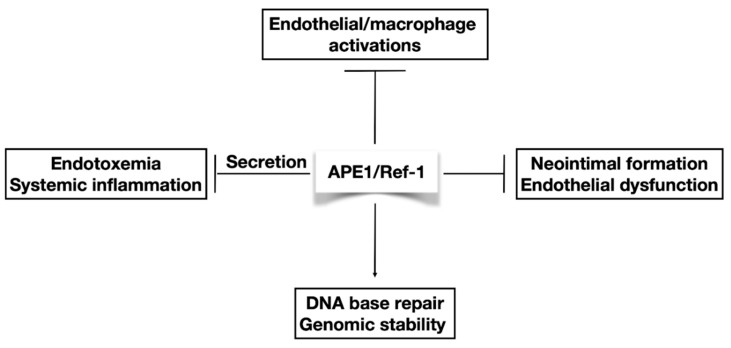
The role of apurinic/apyrimidinic endonuclease 1/redox factor-1 (APE1/Ref-1) in endothelial activation and systemic inflammation. Heterozygous APE1/Ref-1 mice showed endothelial dysfunction and hypertension [19]; gene transfer of APE1/Ref-1 inhibited neointimal formation of rat carotid arteries and inhibited endothelial activation in endothelial cells [18,23]. The secretory APE1/Ref-1 inhibited proinflammatory cytokines and inflammation in lipopolysaccharide-treated mice [22]. APE1/Ref-1 functions in DNA base repair and is essential for genomic stability. The arrow and T-bar represent activated and inhibitory interactions, respectively.

**Figure 2 biomedicines-08-00057-f002:**
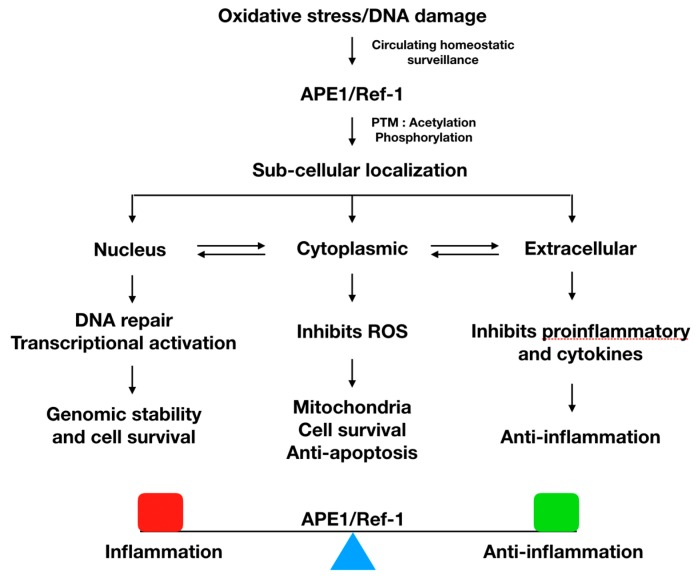
Flowchart model of APE1/Ref-1 and its subcellular localization and functions in response to oxidative stress and DNA damage. APE1/Ref-1 carries out circulating homeostatic surveillance in the human body by recognizing the cellular changes in response to oxidative stress or DNA damage. Subcellular localization of APE1/Ref-1 can be determined by post-translational modification including redox change, acetylation, phosphorylation, nitrosation, etc. Overall, APE1/Ref-1 is involved in DNA base repair and the modulating transcriptional factors, resulting in genomic stability or cell survival. Under basal conditions, APE1/Ref-1 is localized in the nucleus; its localization is dynamically regulated, which results in cytoplasmic/mitochondrial translocation or extracellular secretion.

**Figure 3 biomedicines-08-00057-f003:**
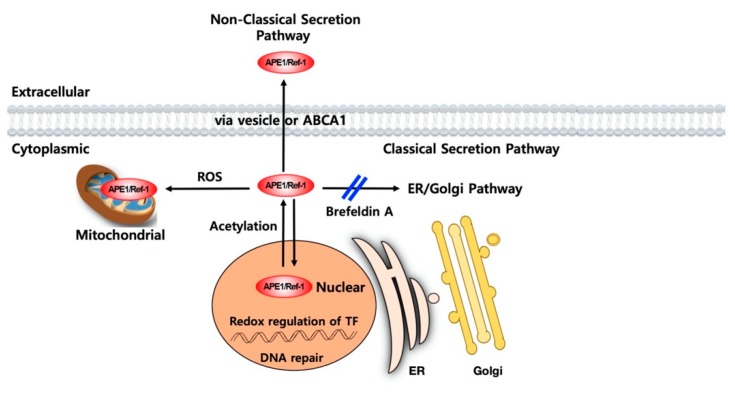
Proposed secretion mechanism of APE1/Ref-1. APE1/Ref-1 is mainly localized in the nucleus, which is dynamically regulated between the cytoplasm or mitochondria. Also, APE1/Ref-1 may be secreted in response to acetylation. Its secretion is not inhibited by brefeldin A, an inhibitor of the ER/Golgi pathway (‘double slash’ in blue), suggesting a nonclassical secretion pathway. Active secretion of APE1/Ref-1 may be mediated by the ABCA1 transporter or vesicle formation [54,57].

**Table 1 biomedicines-08-00057-t001:** Summary of functions of APE1/Ref-1 according to subcellular localization.

APE1/Ref-1	Tissue/Cells	Functions	Reference
Intracellular	Endothelial cells	Inhibits endothelial dysfunction Inhibits cellular ROS and increases NO production Inhibits NF-kB and apoptosis Inhibits VCAM-1 expression	[18,19,23,29,50,51]
	Endothelial mitochondria	Inhibit mitochondrial dysfunctions Inhibits mitochondrial ROS Inhibits p66shc activation Mitochondrial DNA repair	[41,52] [42]
	A549 cells	Inhibits COX-2 expression Inhibits p38 MAPK	[53]
	Vascular smooth muscle cells	Inhibits Pi-induced calcification Inhibits osteoblastic phenotype changes	[31]
Cytoplasmic	Endothelial cells	Inhibits rac1 or NADPH oxidase	[29,39]
	Glial cells	Inhibits neuroinflammatory response	[37]
	THP-1 cells	Inhibits HMGB1-mediated ROS and cytokines	[35]
Extracellular	HEK293 cells	Trichostatin A induced APE1/Ref-1 secretion	[44]
	MDA-MD-231 cells	Ac-APE1/Ref-1 induces apoptosis	[54]
	Endothelial cells	Inhibits VCAM-1 expression	[48]
		Inhibits COX-2 expression	[22]

**Table 2 biomedicines-08-00057-t002:** Summary of usefulness of APE1/Ref-1 as potential biomarker in vascular inflammatory disease or tumors.

Diseases	Clinical Significance	Patients (n)	Control (n)	Sensitivity (%)	Specificity (%)	AUC or 95% CI	Reference
Liposaccharide-induced endotoxemia (Preclinical study)	APE1/Ref-1 is elevated in plasma of lipopolysaccharide (LPS)-treated mice and reached a maximum at 12 h after injection of LPS.	-	-	NA	NA		[45]
Viral myocarditis (Preclinical study)	Serum APE1/Ref-1 is increased in coxsackievirus-induced myocarditis and is well-correlated with the degree of myocardial inflammation. Serum APE1/Ref-1 is useful for myocardial injury in viral myocarditis without endomyocardial biopsy	-	-	NA	NA		[55]
Coronary arterial diseases	Serum APE1/Ref-1 level was higher in coronary arterial diseases, which higher in myocardial infarction than angina	303	57	36	95	0.66	[61]
Bladder cancer	Urinary APE1/Ref-1 is increased in bladder cancer and it correlated with tumor grade and stage	169	108	82	80	0.83	[67]
Oral squamous cell carcinoma	Serum APE1/Ref-1 is a novel potential diagnostic biomarker of oral cancer and can reflect the treatment responses	58	40	67	87	0.80	[74]
Cholangiocarcinoma	Serum APE1/Ref-1 level is a potential diagnostic marker of cholangiocarcinoma and cytoplasmic expression in cancer cells predicts relapses	46	39	73.9	97.4	0.709–0.886	[71]
Hepatocellular carcinoma	Serum APE1/Ref-1 may be considered as a promising diagnostic biomarker for hepatocellular carcinoma	99	100	98	83	0.98	[69]
Renal cell carcinoma	Serum APE1/Ref-1 level may be a diagnostic markers of renal cell carcinoma	40	39	82.5	97.4	0.862–0.981	[70]
Non-small cell lung cancer	Serum APE1/Ref-1 is a biomarker for predicting prognosis and therapeutic efficacy in nonsmall cell lung cancer and post-treatment high serum APE1/Ref-1 level was associated with poor survival.	200	200	55.6	70.8	0.653	[72]
Gastric cancer (lymph node positive and negative	Serum APE1/Ref-1 is a valuable marker for prediction of lymph node metastasis in patients with gastric cancer	52	35	49	85.7	0.666	[73]

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
