# Peer review of "The Biological Role of Apurinic/Apyrimidinic Endonuclease1/Redox Factor-1 as a Therapeutic Target for Vascular Inflammation and as a Serologic Biomarker"

_biomedicines, 2020, doi:10.3390/biomedicines8030057_

Round 1

Reviewer 1 Report

To:

Editorial Board

Biomedicines

Title: “The biological role of apurinic/apyrimidinic endonuclease1/redox factor-1 as a therapeutic target for vascular inflammation and as a serologic biomarker”

Dear Editor,

I read this paper and I think that:

The English of the paper should be revised due to typos. Please revise. The authors should include a table gathering the main findings from clinical studies and clinical application of such a biomarker. Please provide. The reproducibility of the evaluation of such biomarker should be better expressed. The exact role of this biomarkers in each clinical setting (CAD, cancer, etc) should be better described and differentiate. Furthermore, future perspectives should be better outlined.

Author Response

I read this paper and I think that:

The English of the paper should be revised due to typos. Please revise. The authors should include a table gathering the main findings from clinical studies and clinical application of such a biomarker. Please provide. The reproducibility of the evaluation of such biomarker should be better expressed. The exact role of this biomarkers in each clinical setting (CAD, cancer, etc) should be better described and differentiate. Furthermore, future perspectives should be better outlined.

--> Thank for comments. Reviewer comments would be helpful to show the functional role and clinical significance of APE1/Ref-1. The clinical biomarker and its application studies of recent studies and role of this biomarker are presented as the addition of Table 2 of revised manuscript. Clinical significance in vascular inflammatory disease and tumors and its study scale is provided in Table 2. Most of them were small scale studies, and future comprehensive studies on the clinical significance with large scale are needed and also proteomic analysis of post-translational modification of APE1/Ref-1 in biological samples would improve the understanding of the diversity of APE1/Ref-1 function. In overall, typographical error and grammar were re-checked, also missing references were added or re-checked in revised version. In terms of other reviewers' comment, Figure 3 was changed into graphical figure for proposed secretion pathway of APE1/Ref-1.

Reviewer 2 Report

In this review, Lee and collaborators explored the role of APE1/Ref-1 in several cellular processes, pointing evidences towards the dual role of that these proteins have on inflammation scenarios. They also gathered information about APE1/Ref-1 cellular localization in different conditions and the processes that trigger APE1/Ref-1 release to the extracellular milieu. Interestingly, the authors also suggest that APE1/Ref-1 can be used as a biomarker for multiple conditions, including coronary artery disease, myocarditis and cancer. Even though the manuscript contributes to the general knowledge about APE1/Ref-1, there are some issues that the authors need to address before this paper can be considered for publication. The minor and major concerns are listed below:

MINOR:

Line 27: The word “inflammation” does not require a capital letter in the beginning.

Line 48: The word “secretion” in this case does not go to the plural form.

Line 72: AS well as.

Figure 1, line 70: Remove the plural of “inflammation”.

Line 106: The authors should define the acronym HMGB-1 the first time they use it in the text.

Figure 2: Please, substitute the word “survivals” for “survival” within the figure.

Table 1: So far TSA had not been introduced in the manuscript. Please make sure to define the acronym before using it in the text.

Table 1: Please keep the table format consistent. Mice/ Endothelial cells (in vivo), following what the authors have done for rat endothelial cells above on the same table.

MAJOR:

Lines 32-33: What do the authors mean by "host cells" in this case? Inflammatory cells also belong to the "host". Please clarify or change the wording.

Lines 53-55: Please improve the link of ideas between lines 53 and 54. What is the connection between cytokine release and intracellular cholesterol trafficking?

Line 59: Please rephrase the title to convey the information contained in the section that it refers to. E.g.: "APE/Ref-1 proteins have several cellular functions".

Line 70: “It was found that designated secretory APE1/Ref-1 inhibited proinflammatory cytokines and the inflammation in lipopolysaccharide-treated mice”. What do the authors mean by “designated secretory”? Please consider rephrasing this to “It was found that the secreted APE1/Ref-1…”.

Figure 1: The figure caption for figure 1 needs to be significantly improved in a way that the reader is able to understand, just by looking at the figure and its respective caption, how secreted APE/Ref-1 is involved in these 4 different pathways that are depicted. Also, make sure to clarify the "LPS-induced septic mouse figure" on the left (what is it showing to the reader, the heat maps and whatnot) and the graph on the right (axes and meaning of these curves).

Lines 207-209: Please add references here that can support the authors’ claim that there is a “passive secretion” of APE7Ref-1 to the extracellular milieu. I think that “passive secretion” is not a valid term here since it is not, as far as I am concerned, based on any gradient concentration (going pro or against it). I would probably use "release" or a similar word. In this case, because the nuclear membranes might be ruptured due to cell death processes, the APE/Ref-1 is leaking from the nucleus.

“In necrotic or apoptotic cells, APE1/Ref-1 could be released in the cultured medium from nucleus.” Since there is a direct release of APE1/Ref-1 from the nucleus to the extracellular milieu, the authors should add double arrows on Fig 2 connecting the nucleus to the extracellular milieu as well.

Figure 3:

1) Why would the red curve on the normal range go below the normal (basal) level? Shouldn't they be around at the same level, parallel to the x-axis)?

2) Is the measured amount of APE17Ref-1 in the serum higher in the case of passive release? Please explain this further (with references).

Author Response

Reviewer #2

In this review, Lee and collaborators explored the role of APE1/Ref-1 in several cellular processes, pointing evidences towards the dual role of that these proteins have on inflammation scenarios. They also gathered information about APE1/Ref-1 cellular localization in different conditions and the processes that trigger APE1/Ref-1 release to the extracellular milieu. Interestingly, the authors also suggest that APE1/Ref-1 can be used as a biomarker for multiple conditions, including coronary artery disease, myocarditis and cancer. Even though the manuscript contributes to the general knowledge about APE1/Ref-1, there are some issues that the authors need to address before this paper can be considered for publication. The minor and major concerns are listed below:

MINOR:

Line 27: The word “inflammation” does not require a capital letter in the beginning.

--> Thanks for detail comment, we fixed it.

Line 48: The word “secretion” in this case does not go to the plural form.

--> Thanks for detail comment, we fixed it.

Line 72: AS well as.

--> Thanks for detail comment, we fixed it.

Figure 1, line 70: Remove the plural of “inflammation”.

--> Thanks for detail comment, we fixed it.

Line 106: The authors should define the acronym HMGB-1 the first time they use it in the text.

--> Thanks for detail comment, we fixed it like “high mobility group box-1 (HMGB-1)”

Figure 2: Please, substitute the word “survivals” for “survival” within the figure.

--> Thanks for detail comment, we fixed it.

Table 1: So far TSA had not been introduced in the manuscript. Please make sure to define the acronym before using it in the text.

--> Thanks for detail comment, we fixed it like “trichostatin A”

Table 1: Please keep the table format consistent. Mice/ Endothelial cells (in vivo), following what the authors have done for rat endothelial cells above on the same table.

--> Thanks, we re-adjusted the contents of the table to make it consistent. Also, in term of other reviewers' concerns, usefulness of APE1/Ref-1 as potential biomarker was provided as the addition of new Table 2.

MAJOR:

Lines 32-33: What do the authors mean by "host cells" in this case? Inflammatory cells also belong to the "host". Please clarify or change the wording.

--> Thanks for detail comment. To clarify, we changed “defense cells” instead of “host cells”

Lines 53-55: Please improve the link of ideas between lines 53 and 54. What is the connection between cytokine release and intracellular cholesterol trafficking?

--> Thank for comment, we added the connection sentence in line 55 like "Sine cholesterol is a key component of arterial plaques and the understanding of cholesterol transport system help to lower the risk of atherosclerosis"

Line 59: Please rephrase the title to convey the information contained in the section that it refers to. E.g.: "APE/Ref-1 proteins have several cellular functions".

--> Thanks for comment. we rephrase “APE1/Ref-1 protein have several cellular functions” in line 66.

Line 70: “It was found that designated secretory APE1/Ref-1 inhibited proinflammatory cytokines and the inflammation in lipopolysaccharide-treated mice”. What do the authors mean by “designated secretory”? Please consider rephrasing this to “It was found that the secreted APE1/Ref-1…”.

--> Thanks for detail comment, we fixed it as you comment in line 79.

Figure 1: The figure caption for figure 1 needs to be significantly improved in a way that the reader is able to understand, just by looking at the figure and its respective caption, how secreted APE/Ref-1 is involved in these 4 different pathways that are depicted.

--> Thanks for valuable comment. Figure 1 has been significantly modified to make it easier to understand. Also, the potential secretion pathway of APE1/Ref-1 are presented, based on recent results in Figure 3. 

Also, make sure to clarify the "LPS-induced septic mouse figure" on the left (what is it showing to the reader, the heat maps and whatnot) and the graph on the right (axes and meaning of these curves).

--> To avoid confusion, inlet figure in the lower left and right corner has been removed.

Lines 207-209: Please add references here that can support the authors’ claim that there is a “passive secretion” of APE7Ref-1 to the extracellular milieu. I think that “passive secretion” is not a valid term here since it is not, as far as I am concerned, based on any gradient concentration (going pro or against it). I would probably use "release" or a similar word. In this case, because the nuclear membranes might be ruptured due to cell death processes, the APE/Ref-1 is leaking from the nucleus.

--> Thanks for nice comment, I can agree your opinion about passive secretion. “Passive release” seems to be more appropriate. Like HMGB-1, APE1/Ref-1 also could be released from cell damage or necrotic cells. In general, like your opinion, APE1/Ref-1 is leaking to extracellular milieu from cell, due to cell membrane or nuclear membrane rupture during cell death process. 

“In necrotic or apoptotic cells, APE1/Ref-1 could be released in the cultured medium from nucleus.” Since there is a direct release of APE1/Ref-1 from the nucleus to the extracellular milieu, the authors should add double arrows on Fig 2 connecting the nucleus to the extracellular milieu as well.

--> Thank for good comment. We thought that APE1/Ref-1 released from the nucleus will release to extracellular milieu after reaching the cytoplasm during cell death or necrosis, rather than directly released in the culture medium from nucleus. Therefore, we fixed sentence like “APE1/Ref-1 could be released in the cultured medium form cytoplasm or nucleus.” in line 217.  

Figure 3:

1) Why would the red curve on the normal range go below the normal (basal) level? Shouldn't they be around at the same level, parallel to the x-axis)?

--> Thanks for question. Figure 3 showed the author’s hypothetical concept idea. It is expected to be clear if direct experimental evidence or study is available. Normal value can be defined as basal APE1/Ref-1 level in healthy persons. Because the cells have normal metabolism and activity, zero values, “parallel to the X-axis” are not expected. Anyway, to avoid confusion and no direct evidence about this concept, Figure 3 is removed and replaced by the potential APE1/Ref-1 secretion pathway as you comment in early.

2) Is the measured amount of APE17Ref-1 in the serum higher in the case of passive release? Please explain this further (with references).

--> Even there is no direct clinical reference about APE1/Ref-1 amount is higher in case of passive releases compared with active secretion. In vitro experiment, the active secreted APE1/Ref-1 level, such as caused by hyperacetylation, is lesser than passive release, such as caused by cell apoptosis. Furthermore, there is also interesting paper about serum APE1/Ref-1 level is higher in hepatic diseases caused by hepatitis B virus, compared with liver tumors such as hepatocellular carcinoma or cholangiocarcinoma, suggesting higher APE1/Ref-1 level in passive release by cell/tissue injury. (Kim JM et al, 2019, Journal of Clinical Medicine). Anyway, to avoid confusion and no direct evidence about this concept, Figure 3 is removed and replaced by the potential APE1/Ref-1 secretion pathway as you comment in early.

Reviewer 3 Report

The AA described the possible biological functions of APE1/Ref-1 according to its subcellular localization or its extracellular secretions, and also as therapeutic targets for vascular inflammation.

No concerns or other  improvements are needed for this overview.

Author Response

The AA described the possible biological functions of APE1/Ref-1 according to its subcellular localization or its extracellular secretions, and also as therapeutic targets for vascular inflammation. No concerns or other improvements are needed for this overview.

--> Thanks for positive review. In overall, typographical error and grammar were re-checked, also missing references were added or re-checked in revised version. In terms of other reviewers’ comment, Figure 3 was changed into graphical figure for proposed secretion pathway of APE1/Ref-1.

Reviewer 4 Report

Manuscript written by Yu Ran Lee et. al., is a review article on the role of APE1/Ref-1 in vascular inflammation and its use as a diagnostic biomarker.

Manuscript is scientifically sound and coherent. However, manuscript still has some concerns with regard to text connectivity and missing reference numbers . 

Major concerns;

1) In first section of manuscript, author's have suggested various targets for atherosclerosis such as " anti-adhesion therapy" , "blocking of cytokines"  and "cholesterol export". However, author's have failed to bring rationale for the need for new targets. Whole paragraph needs to be written more precisely with better connectivity and at the end APE1 should be introduced .

2) At page 2 of 12, line 63, Author says "Here? we introduce the biological functions of APE1?Ref?1?" must be removed since this is not the first paper reports about APE1/Ref-1. or this should be written more appropriately with author's own citation.

3) At page2 of 12, line 68, author talks about heterozygous APE1/Ref-1 knock out mice. Is there anything known about homozygous APE1/Ref-1 knock out mice? Author should include information about homozygous APE1/Ref-1 mice.

4) In Figure 3; Y axis does not show any values in ng/ml. Author must mention actual numbers in ng/ml based on earlier reports. 

5) Extracellular APE1/Ref-1 is an anti-inflammatory. Is APE1 secreted during diseased condition is also an anti-inflammatory? 

6) Throughout the manuscript authors have missed citing references. Each specific statement must have its own reference. Some of the line missing reference number are; line 31, 33, 35, 39, 42,45, 46, 48, 55, 67.... Please check entire manuscript for this. 

Minor concern;

Page 7 of 12, line 220; Spelling mistake: myocadiac.

Author Response

Reviewers #4

Manuscript written by Yu Ran Lee et. al., is a review article on the role of APE1/Ref-1 in vascular inflammation and its use as a diagnostic biomarker. Manuscript is scientifically sound and coherent. However, manuscript still has some concerns with regard to text connectivity and missing reference numbers.

Major concerns;

In first section of manuscript, author's have suggested various targets for atherosclerosis such as " anti-adhesion therapy" , "blocking of cytokines"  and "cholesterol export". However, author's have failed to bring rationale for the need for new targets. Whole paragraph needs to be written more precisely with better connectivity and at the end APE1 should be introduced.

--> Thanks for nice comment.  We added the end of introduction for better connectivity for the need for new targets. “A new target molecule capable of efficiently monitoring vascular inflammation, extracellularly secreted as needed to act as a biomarker, and able to control vascular inflammation including sepsis or cytokine storms, is required. Here, we introduce APE1/Ref-1 as a potential new target capable of meeting these demands”.  

2) At page 2 of 12, line 63, Author says "Here? we introduce the biological functions of APE1?Ref?1?" must be removed since this is not the first paper reports about APE1/Ref-1. or this should be written more appropriately with author's own citation.

--> Thanks for comment. We can agree to your opinion. We removed “Here we introduce the biological function of APE1/Ref-1” since it is not the first paper report about APE1/Ref-1.

3) At page2 of 12, line 68, author talks about heterozygous APE1/Ref-1 knock out mice. Is there anything known about homozygous APE1/Ref-1 knock out mice? Author should include information about homozygous APE1/Ref-1 mice.

--> Thanks for comment. The importance of APE1/Ref-1 is known by the fact that homozygotic knockout of APE1/Ref-1 cannot survive. The conventional knockout of APE1/Ref-1 causes early embryonic lethality on embryonic days E5 to E9 (Reference 20,21). Therefore, it is difficult to study the biological functions of APE1/Ref-1 in homozygous APE1/Ref-1 knockout mice. We provide the information about homozygous APE1/Ref-1 mice in line 77 of revised version. 

Reference:

  1. Ludwig DL, MacInnes MA, Takiguchi Y, Purtymun PE, Henrie M, Flannery M, Meneses J, Pedersen RA, Chen DJ. A murine AP-endonuclease gene-targeted deficiency with post-implantation embryonic progression and ionizing radiation sensitivity Mutat Res. 1998 Oct 21;409(1):17-29. PMID 9806499)
  2. Xanthoudakis S, Smeyne RJ, Wallace JD, Curran T. The redox/DNA repair protein, Ref-1, is essential for early embryonic development in mice. Proc Natl Acad Sci U S A. 1996 Aug 20;93(17):8919-23. PMID 8799128)

4) In Figure 3; Y axis does not show any values in ng/ml. Author must mention actual numbers in ng/ml based on earlier reports.

--> Thanks for question.  Figure 3 shows the author’s hypothetical concept idea. Based on the research paper to date, it is still difficult to show the actual value. It is expected to be clear if direct experimental evidence or study is available. Anyway, to avoid confusion and no direct evidence about this concept, Figure 3 is removed and replaced by the potential APE1/Ref-1 secretion pathway.

5) Extracellular APE1/Ref-1 is an anti-inflammatory. Is APE1 secreted during diseased condition is also an anti-inflammatory?

--> As described in the section 2-3, the adenoviral secretory APE1/Ref-1 or recombinant APE1/Ref-1 showed anti-inflammatory activity. Joo et al reported that the secreted APE1/Ref-1 in TNF-treated HUVEC or LPS-treated macrophage showed anti-inflammatory activity because anti-inflammatory activity of secreted Ref-1 is inhibited by neutralization with specific antibody against Ref-1. However, it is not confirmed in vivo condition whether secreted APE1/Ref-1 showed anti-inflammatory action in certain diseases. In order to confirm this, it is necessary to investigate the effect of neutralization of secreted APE1/Ref-1 using specific antibody in vivo.

6) Throughout the manuscript authors have missed citing references. Each specific statement must have its own reference. Some of the line missing reference number are; line 31, 33, 35, 39, 42,45, 46, 48, 55, 67.... Please check entire manuscript for this.

--> Thanks, we re-checked missing references in entire manuscript. Total 19 references were added in revised version. 

Minor concern;

Page 7 of 12, line 220; Spelling mistake: myocadiac.

--> Thanks for detail comment, we fixed it.

Round 2

Reviewer 1 Report

To:

Editorial Board

Biomedicines

Title: “The biological role of apurinic/apyrimidinic endonuclease1/redox factor-1 as a therapeutic target for vascular inflammation and as a serologic biomarker”

Dear Editor,

I read the revised version of this paper and I think that the authors well addressed my previous comments. The paper improved very much.

Reviewer 2 Report

Since the authors addressed most of my concerns I can recommend the paper to be accepted in the present form.

Just one comment to the authors: whenever you make changes on your manuscript that were requested by the reviewers, please put the line numbers of the modified/updated version of the paper on the rebuttal letter so it is easier to track where the suggestions were implemented.